# Hunger for Home Delivery: Cross-Sectional Analysis of the Nutritional Quality of Complete Menus on an Online Food Delivery Platform in Australia

**DOI:** 10.3390/nu13030905

**Published:** 2021-03-11

**Authors:** Celina Wang, Andriana Korai, Si Si Jia, Margaret Allman-Farinelli, Virginia Chan, Rajshri Roy, Rebecca Raeside, Philayrath Phongsavan, Julie Redfern, Alice A. Gibson, Stephanie R. Partridge

**Affiliations:** 1Nutrition and Dietetics Group, School of Life and Environmental Science, Charles Perkins Centre, The University of Sydney, Sydney, NSW 2006, Australia; margaret.allman-farinelli@sydney.edu.au (M.A.-F.); vcha3326@uni.sydney.edu.au (V.C.); 2Westmead Applied Research Centre, Faculty of Medicine and Health, The University of Sydney, Sydney, NSW 2145, Australia; sisi.jia@sydney.edu.au (S.S.J.); rebecca.raeside@sydney.edu.au (R.R.); julie.redfern@sydney.edu.au (J.R.); stephanie.partridge@sydney.edu.au (S.R.P.); 3Discipline of Nutrition and Dietetics, Faculty of Medical and Health Sciences, The University of Auckland, Auckland 1011, New Zealand; r.roy@auckland.ac.nz; 4Prevention Research Collaboration, Sydney School of Public Health, The University of Sydney, Sydney, NSW 2006, Australia; philayrath.phongsavan@sydney.edu.au; 5The George Institute for Global Health, The University of New South Wales, Camperdown, NSW 2006, Australia; 6Menzies Centre for Health Policy, Sydney School of Public Health, Faculty of Medicine and Health, The University of Sydney, Sydney, NSW 2006, Australia; alice.gibson@sydney.edu.au

**Keywords:** food environment, online food delivery, independent outlet, takeaway foods, fast food, nutrition, meal deals, adolescent, young adult

## Abstract

Online food delivery (OFD) platforms have changed how consumers purchase food prepared outside of home by capitalising on convenience and smartphone technology. Independent food outlets encompass a substantial proportion of partnering outlets, but their offerings’ nutritional quality is understudied. Little is also known as to how OFD platforms influence consumer choice. This study evaluated the nutritional quality and marketing attributes of offerings from independent takeaway outlets available on Sydney’s market-leading OFD platform (UberEats^®^). Complete menus and marketing attributes from 202 popular outlets were collected using web scraping. All 13841 menu items were classified into 38 food and beverage categories based on the Australian Dietary Guidelines. Of complete menus, 80.5% (11,139/13,841) were discretionary and 42.3% (5849/13,841) were discretionary cereal-based mixed meals, the largest of the 38 categories. Discretionary menu items were more likely to be categorised as most popular (OR: 2.5, 95% CI 1.9–3.2), accompanied by an image (OR: 1.3, 95% CI 1.2–1.5) and offered as a value bundle (OR: 6.5, 95% CI 4.8–8.9). Two of the three discretionary food categories were more expensive than their healthier Five Food Group counterparts (*p* < 0.02). The ubiquity of discretionary choices offered by independent takeaways and the marketing attributes employed by OFD platforms has implications for public health policy. Further research on the contribution of discretionary choices and marketing attributes to nutritional intakes is warranted.

## 1. Introduction 

The prevalence of consuming food prepared out of home, including takeaway stores, fast food chains, and restaurants, has increased globally due to busier lifestyles and demand for convenience [1,2]. Almost 50% of global respondents to a market research survey eat away from home at least once a week [3]. In Australia since the late 1980s, the proportion of total food expenditure on foods prepared outside of home has increased from 25% to 34% [4]. However, frequent consumption of takeaway food has been reported to be associated with poorer diet quality, namely, high levels of energy, total fat and sodium intake, and higher prevalence of obesity [2,5,6,7].

The physical, economic, political, and socio-cultural context in which consumers engage with the food system to make their decisions about acquiring, preparing, and consuming food is known as the food environment [8]. The existing food environment is being disrupted by the emergence of online food delivery (OFD) platforms, which have recorded a doubling in usage from 2018 to 2020 [9]. OFD platforms have been defined as “websites or smartphone applications set up for customers to select from food outlets’ menu items to order food prepared away from the home for pick up or delivery by freelance couriers” [10]. Almost 30% of younger generations (15-to-34 years) use these OFD platforms and have recently been identified as their main users [11]. Food prepared outside of home has recently been reported to be most consumed by young people compared to other age groups [12]. Young people have also been reported to spend just under $AUD 2000 per year on these foods [4].

Concerns have been raised about the nutritional quality of meals offered by OFD platforms. A recent cross-sectional observational study conducted by Partridge et al. in Sydney and Auckland, characterised 680 popular food outlets and their five to ten most popular menu items from UberEats^®^, the market-leading OFD platform in Australia [10]. They found 86% of popular menu items were discretionary foods and beverages (also known as junk food), which are defined as items high in added salt, saturated fat, sugar and low in fibre by the Australian Dietary Guidelines [10,13]. Previous studies have described the nutritional composition of meals from takeaway franchises (chain stores that prepare and sell meals/snacks ready for immediate consumption, offered in specialised packaging, e.g., McDonalds^®^, or KFC^®^) that are subject to menu labelling [14,15]. However, independent takeaways (takeaway outlets that are not franchises, e.g., local kebab shop) [16], are not subject to this regulation and were reported to be the second largest (30%) food outlet type on OFD platforms after takeaway franchises [10]. Only a few studies have explored the nutritional quality of food items provided by independent outlets [5,10,17], highlighting a research gap exists for this food outlet classification. The collation of independent restaurants on a single platform provides a unique opportunity for efficient evaluation of the nutritional quality of menus from multiple independent takeaways.

OFD platforms have also revolutionised the way we purchase food prepared out of home by capitalising on smartphone technology and subsequently expanding our food choices beyond local food outlets [18,19]. The platforms allow users to order food, view enticing food images, and access other customers’ reviews with an unprecedented ease [19]. A recent Australian study reported price, value for money food items, and appealing food images to heavily influence young people’s preference to eat food prepared outside of home [20]. 

Marketing tactics that position foods to be first viewed and default popular choices may encourage their selection. Within an online grocery store context, options placed on the “first-screen” appear to be more commonly selected [21]. Such strategies are akin to the positioning of food items on supermarket shelves at eye-level and at check-out counters [22]. The position of an item within a physical menu and the use of popularity cues such as “Most Popular” have also been reported to sway consumer choice [23,24]. Popularity cues serve as an indicator of product demand and/or interest by other consumers [24]. 

Nutritional labelling at point-of-sale may also serve as a form of marketing to encourage choice, or in this case to guide consumers to healthier choices [25,26,27]. A study conducted at a New Zealand university also reported use of symbols on menus to label healthy foods influenced food selection [28]. 

The use of marketing techniques within the unique digital food environment of OFD platforms and the association with nutritional quality of menus offered warrants further investigation.

Thus, this study’s primary aim was to evaluate the nutritional quality of complete menus from popular independent takeaways available on a market-leading OFD platform in areas with high proportions of young consumers (15–34-years) in Australia. The secondary aim was to investigate the associations between the nutritional quality and marketing attributes of these menus, including a popularity cue, image use, prices, offerings as value bundles, nutritional information, and dietary labelling. 

## 2. Materials and Methods 

### 2.1. Identification of Popular Independent Takeaways

This study formed a new database of complete menus for food outlets that were identified by a previous cross-sectional study conducted in Sydney between 9 and 22 February 2020. The identification process is described in detail elsewhere [10]. The 10 most popular food outlets were extracted from the UberEats’ “popular near you” section for areas with above-average populations of young people (>30% 15–34 years), the leading users of OFD platforms [10]. Researchers were not logged into personal UberEats accounts to avoid biased results. This project focused on evaluation of the 202 independent takeaways identified from the previous study using the Food Environment Score Tool [16,29,30]. Independent takeaways have been defined as outlets that are not franchises, which prepare and sell meals or snacks, ready for immediate consumption, e.g., local kebab shop [16].

### 2.2. Data Extraction

Publicly available complete menus were extracted from the OFD website on September 10th 2020 (via web scraping, ScrapingSolutions). Complete menus include all menu items available from independent takeaways as displayed on their UberEats webpage. Data collected from these menus included the menu items’ names, descriptions, UberEats categories, prices, images, nutritional information (e.g., the macronutrient profile), and any dietary labelling (e.g., vegan, vegetarian, or gluten-free). Table 1 provides a summary of definitions and derivations of the data extracted. 

### 2.3. Outcome Measures

The primary outcome of this study was the nutritional quality of complete menus from popular independent takeaways. Secondary outcomes were the prevalence of marketing attributes for complete menus and their association with nutritional quality.

#### 2.3.1. Nutritional Quality

All menu items from independent takeaways were classified into 38 food and beverage categories using a modified version of a classification system previously proposed for a sub-study of the existing MYMeals project (Appendix A) [7]. These food and beverage categories define menu items using food types derived from the Australian Dietary Guidelines’ Five Food Group (FFG) and discretionary classifications [13]. FFG dishes (also known as core) contain food(s) or a combination of foods from the five food groups: vegetables and legumes/beans; fruit, grain (cereal) foods, mostly wholegrain, and/or high cereal fibre varieties; lean meats and poultry, fish, eggs, tofu, nuts and seeds, and legumes/beans; and milk, yoghurt, cheese and/or alternatives, and mostly reduced fat) [10,13]. Discretionary dishes are defined as items high in added salt, saturated fat, sugar, and low in fibre by the Australian Dietary Guidelines [10,13]. Some menu items lacked substantial data and were classified as “undetermined” as there were multiple categories to which assignment was possible (e.g., “drink” with no image or description provided). Other menu items were not edible (e.g., cutlery, or napkins) and classified as “non-consumable”. Both the undetermined and non-consumable categories were excluded from data analysis. Where a menu item was not considered in the original classification system, the Australian Bureau of Statistics’ (ABS) principles and list for identifying discretionary foods were used to assist categorisation [31]. A conservative approach in favour of FFG was taken when insufficient information was available to classify menu items as discretionary. For example, stir-fries without adequate detail in the description were classified as a FFG type dish although some stir-fries have excessive sodium. The database was sorted alphabetically by the UberEats food category and then by menu item name. Two dietetic researchers each classified half the database (AK, CW). A random 20% sample of the data was cross-checked by Accredited Practising Dietitians (SRP, SJ) with an agreement of 99.0% in the cross-checked sample.

#### 2.3.2. Marketing Attributes 

Marketing attributes included: popularity cue (category of “Most Popular”), price, value bundles, use of image, nutritional information, and dietary labelling. The data required to assess all marketing attributes, excluding value bundles, was extracted by the web scraping company. Value bundles such as catering and party packs, meal deals, and family deals were manually coded using the menu item name and description. The higher cost of value bundles was anticipated to inflate the median price and thus excluded from price analysis. Table 1 provides a summary of the definitions of these study outcomes. 

### 2.4. Data Analysis

All data was collated on Microsoft Excel (Version 16.41, Microsoft Corporation, Redmond, Washington, DC, USA). Any food and beverage categories which were less than 10% of all menu items were grouped into four categories: Other Food (discretionary), Other Food (FFG), Other Beverage (discretionary), and Other Beverage (FFG). Descriptive statistics were used to evaluate the nutritional quality and marketing attributes of all menu items. Categorical variables (nutritional quality, popularity, value bundles, image, nutritional information and dietary labelling) were summarised using frequencies and proportions. Chi-squared tests with the Bonferroni multiple comparisons correction and odds ratios were used for categorical variables to identify significant differences between (i) discretionary and FFG menu items and (ii) most popular and regular menu items. The distribution of continuous variables (price) was assessed using histograms and measures of skewness and kurtosis. Continuous variables were summarised as medians and interquartile intervals. Kruskal–Wallis tests with multiple comparisons corrections were used for continuous variables to identify significant differences between (i) most popular and regular menu items and (ii) comparable discretionary and FFG food and beverage categories. All analyses were undertaken using SPSS Statistics Version 26 (IBM, Armonk, New York, NY, USA).

## 3. Results

### 3.1. Selection of Menu Items

A total of 14,103 menu items were available from complete menus of 196 independent takeaways (Figure 1). Menus from six of the independent takeaways were absent from the data extraction. After 262 undetermined, non-consumable and UberEATS category duplicate menu items were excluded, 13,841 menu items remained for analyses of nutritional quality and all marketing attributes, excluding price. Following further exclusion of 1107 value bundles (35 catering or party packs, 395 family deals, and 677 meal deals) 12,734 entries were available for price analysis.

### 3.2. Nutritional Quality and Marketing Attributes

#### 3.2.1. Nutritional Quality and Most Popular Menu Items

The proportions of each food and beverage category for complete menus is depicted in Table 2. The majority of menu items were discretionary (80.5%, 11,139/13,841). The discretionary cereal-based mixed meal category was the largest category within complete menus (42.3%, 5849/13,841) (Table 2). This category included pizzas, burgers, pides, pasta, wraps, sandwiches, and rolls. The second largest category was discretionary meat or alternative-based mixed meals (13.9%, 1924/13,841). This group included items such as Halal snack packs (halal-certified doner kebab meat (e.g., lamb, chicken, or beef) and chips), charcoal chicken, deep-fried chicken, deep-fried seafood meals (e.g., fish and chips), and ribs.

Table 3 depicts the proportion of discretionary and FFG menu items within complete menus and each marketing attribute subgroup. Most popular menu items comprised 4.5% (625/13,841) of complete menus and the majority of the most popular menu items were discretionary (90.8%, 568/625) (Table 3). A discretionary item was more likely (OR: 2.5, 95% CI 1.9–3.2) to be most popular compared to a FFG menu item. The discretionary cereal-based mixed meal category was the largest category from the most popular menu items (57.0%, 357/625) (Appendix A). The second-largest category for most popular menu items was discretionary meat or alternative-based mixed meals (25.2%, 157/625) (Appendix A).

#### 3.2.2. Value Bundles

The frequency of images and meal deals associated with (i) complete menus (*N* = 13,841) and (ii) the most popular menu items (*n* = 625) is depicted in Table 4. Within complete menus, 8.0% (1107/13,841) were a value bundle. A higher proportion of discretionary menu items (9.6%, 1064/11,139) were offered as a value bundle compared to FFG menu items (1.6%, 43/2702) (*p* < 0.001) (Table 4). Discretionary menu items were 6.5 times more likely to be offered as a value bundle than FFG menu items (Table 3). Discretionary cereal-based mixed meals made up 66% (729/1107) of all value bundles (Table 4). There was no significant difference in the number of value bundles within the most popular menu items compared to regular menu items (*p* = 0.549). 

#### 3.2.3. Images

Within complete menus 29.6% (4097/13,841) were accompanied by an image. A higher proportion of discretionary menu items (30.7%, 3420/11,139) had images compared to FFG menu items (25.1%, 677/2702) (*p* < 0.001) (Table 4). Discretionary menu items were 1.3 times more likely to have an image than FFG menu items (Table 3). Most popular menu items were also more likely to have an image compared to the regular menu items (*p* < 0.01).

#### 3.2.4. Price 

Table 5 depicts the prices of the (i) most popular and (ii) regular menu items excluding value bundles. The median price of the most popular items was significantly higher than the regular menu items for discretionary cereal-based mixed meals (*p* = 0.011), meat or alternative-based mixed meals (*p* < 0.001), savoury sauces, condiments, and spreads (*p* = 0.025) and discretionary vegetable-based mixed meals (*p* = 0.021). However, the median price of the most popular menu items for iced confectionary and dairy-based desserts, was significantly less than the regular menu items (*p* = 0.036).

Figure 2 compares the median price between categories with discretionary and FFG counterparts. The median price for discretionary cereal-based mixed meals ($14.00) was higher than its FFG counterpart ($12.00, *p* < 0.001). The median price of discretionary vegetable-based mixed meals ($10.00) was higher than their FFG counterparts ($8.90, *p* = 0.013). However, the median price of discretionary meat or alternative-based mixed meals ($15.50) was lower than its FFG counterpart ($17.80, *p* < 0.001).

#### 3.2.5. Nutritional Information and Dietary Labelling 

Nutritional information was available for 38 menu items and only energy (kJ) values were provided. Dietary labelling was found for 68 menu items, which comprised mostly of vegetarian and vegan labels. As both sample sizes were too small (<1% of all menu items), no data analysis was performed.

## 4. Discussion

To our knowledge, this study is the first to evaluate the nutritional quality of complete menus offered by popular independent takeaways on an Australian market-leading OFD platform and examine the association between their nutritional quality and marketing attributes. Discretionary food and beverages made up the majority of menus and were more likely to be most popular, accompanied by an image and offered as a value bundle than FFG menu items. Nutritional and dietary labelling was also largely absent from our sample. Discretionary mixed meals (cereal-based and vegetable-based) were more expensive than their FFG counterparts. Our findings suggest the menus from independent takeaways are dominated by menu items of poor nutritional quality and these menu items appear to be associated with greater use of marketing attributes. 

Most menu items offered by popular independent takeaways were discretionary. Similar to our finding, Jaworowska et al. examined 489 commonly consumed meals from small, independent takeaway establishments and found they contributed a large proportion of daily energy and salt intake [17,32,33]. Recently, Goffe et al. also reported 79% of 149 takeaway food outlets available on the market-leading OFD platform in England, obtained a health rating score of 2 or less out of a maximum of 5 using a novel outlet-level health metric [34]. Our study also identified the majority of most popular menu items to be discretionary. This affirms Partridge et al.’s finding that 84.3% of most popular menu items were discretionary on the same OFD platform [10]. It is possible that the COVID-19 pandemic may have increased the ordering of unhealthy choices on OFD platforms. A recent observational study in 38 countries reported COVID-19 stay at home policies and COVID-19 induced psychological distress were associated with less preparing and selecting of healthier foods in adults [35]. This is highlighted as UBER reported that compared to the previous year, delivery bookings grew 113% in the second quarter of 2020 and revenue grew 103% in August 2020 [36]. More than half of the most popular menu items were discretionary cereal-based mixed meal dishes such as pizzas and burgers. This finding aligns with a recent cross-sectional study across three international cities, including Melbourne, Australia, which studied outlets on OFD platforms located in different socio-demographic areas. They found that in all three cities, burgers and pizza were the most common predefined keywords used to advertise meals [37]. Thus, the discretionary cereal-based mixed meal dishes may be the most popular menu item of independent takeaways as they are amongst the most visible meals on OFD platforms. However, further research is required to confirm that popularity cues influence OFD users to purchase unhealthy choices. A recurring limitation of prior research was analysis of only the most commonly purchased or consumed menu items available from the food outlets investigated [10,17,33]. By demonstrating that the poor nutritional quality seen in most popular menu items from popular independent takeaways extends to their complete menus, our findings support that in addition to franchise stores, independent outlets should also be included in current public health nutrition discussions.

Discretionary menu items were more likely to be offered as value bundles and accompanied by an image than FFG menu items. A 2017 study in the US reported consumers were more likely to purchase meal deals when presented with the option than when no option was available [38]. This is a public health concern as value bundles add excessive amounts of deleterious nutrients to already nutritionally poor foods [39]. More recently a study focusing on young people living in Australia also reported increased purchasing intention for menu items that suggested value for money such as meal deals [20]. This same study also reported images to be strong persuaders of consumer choice [20]. Leading OFD platforms themselves have also reported the use of appealing food images to boost sales [40,41]. The value of images is reaffirmed by our finding that images appeared significantly more with the most popular menu items than with regular items. Contrary to our findings, a 2020 experimental study in the US found no effect of pictures on influencing purchase intention or spending on a fictious screen, simulating an OFD platform [42]. However, this study was limited to a single menu item, a chicken sandwich, which is not representative of the unhealthier choices found on OFD platforms. The style of image used in this study was also very different to that utilised by UberEats which has specific guidelines for use of images by food outlet partners [41,43]. The use of imagery to market poor nutritional quality food has also been noted on other digital platforms, with a recent study reporting 67% of food images seen by adolescents on social media were of discretionary items [44]. Further research on how images influence the purchasing decisions of a variety of unhealthy choices within OFD platform environments is warranted. Menu labelling has also been reported effective in guiding consumers towards lower caloric options when consuming food prepared out of home [25,26] yet we identified only 0.2% of menu items offered nutritional information. Despite not analysing consumption data, the absence of nutritional labelling, and the greater proportion of discretionary menu items accompanied by marketing attributes, suggests OFD platforms may be unconducive to positive dietary choices.

Unlike other marketing attributes, the effect of price on consumer choice behaviour has been explored within the specific context of OFD platforms. Schulz et al. reported consumers’ exploratory behaviour when selecting a food outlet on a major OFD platform was affected by the outlet’s average price compared to others within the same cuisine [45]. The relationship between price and nutritional quality however on OFD platforms has not been explored. A 2011 study conducted in New Zealand evaluating the availability of healthy choices from 24 fast food stores reported that healthier menu items were cheaper than regular options [14]. Our study reported similar results where menu items from two of the three FFG mixed-meal categories were less expensive than their comparable discretionary category. This contrasts with the common perception that healthy choices are more expensive than unhealthy choices, which act as a barrier for young people to select healthier options [20,46]. Whilst a statistical significance was noted between the lower prices of healthier options than their discretionary counterparts, it is unknown if a price difference of $1–$2 would impact purchasing intention. As taste has been identified as a factor that influences food choices when eating out [20], palatability may be more influential than price. The discretionary meat or alternative-based mixed meal category, which was the second largest discretionary category among the menu items, was cheaper than the FFG counterpart. It is plausible that the FFG meat or alternative-based mixed meals were more expensive as premium, leaner cuts of meat tend to be more expensive than fatty cuts. Ultimately, further research is required to investigate price differences between comparable menu items of opposing nutritional quality on OFD platforms and if this influences consumer choice both independently and in consideration of other platform-related marketing attributes. 

A key strength of this study is the investigation of the largest sample size of complete menus from independent takeways. To our knowledge, only Partridge et al. and Jaworowska et al. have evaluated the nutritional quality of menu items from independent takeaways although were limited to popular menu items only [17,32]. We also employed a comprehensive classification system of 38 food and beverage categories before assigning menu items as discretionary or FFG. This system better defined the offerings from independent takeaways to enhance the understanding of their contribution to the current food environment. Previous research examining the relationship of nutritional quality and food marketing within the digital food environment has focused on online grocery shopping [21]. To our knowledge this is the first study to link the nutritional quality and food marketing attributes for food prepared outside the home within the unique digital environment of OFD platforms. While Partridge et al. [10] demonstrated how OFD platforms increase accessibility to discretionary menu items, this study is one of the first to demonstrate how OFD platforms may also be encouraging the choice of discretionary items over healthier options through marketing tactics. 

However, our findings cannot be generalised to all independent takeaways as we only assessed the most popular independent outlets identified by the preceding study [10]. We also examined the market-leading OFD platform, however there are others with widespread usage in Australia e.g., Menulog and Deliveroo. Thus, we may have excluded popular outlets exclusive to other OFD platforms and their marketing attributes. As we used web scraping for data extraction, we were limited to the marketing attributes available through the web browser interface and not the mobile application interface. We also could not examine promotions and purchase incentives which are only visible on users’ personal accounts. Our findings were interpreted with the assumption that Most Popular referred to items with the greatest sales, however the algorithm used to determine this characteristic is not made publicly available. We also did not examine usage data, thus the associations between nutritional quality and marketing attributes should be interpreted with caution as the findings cannot infer causality. 

## 5. Conclusions

The use of OFD platforms as a method to purchase food prepared outside of home is rising and these platforms hold the potential to influence the nutritional quality of the choices users make. The complete menus of the independent takeaways sampled were reported to consist of predominantly unhealthy choices. This demonstrates the need for this food outlet type to be considered in future public health nutrition policy discussions. As we only examined independent takeaways, future research could examine other types of independent outlets, e.g., independent restaurants. Our nutritional analysis was also limited to food and beverage categorisation because we had no access to the complete recipe data of all menu items. To obtain a more detailed nutrient analysis of independent outlets’ offerings, more detailed food composition data would be needed. The disparity in the use of marketing attributes across discretionary and FFG menu items suggest OFD platforms may be promoting unhealthier options. However, we did not examine consumption nor even sales; thus, the impact remains speculative. We strongly advocate for further research examining how marketing cues in OFD platforms affect nutrition intakes to determine the need to expand menu labelling policies.

## Figures and Tables

**Figure 1 nutrients-13-00905-f001:**
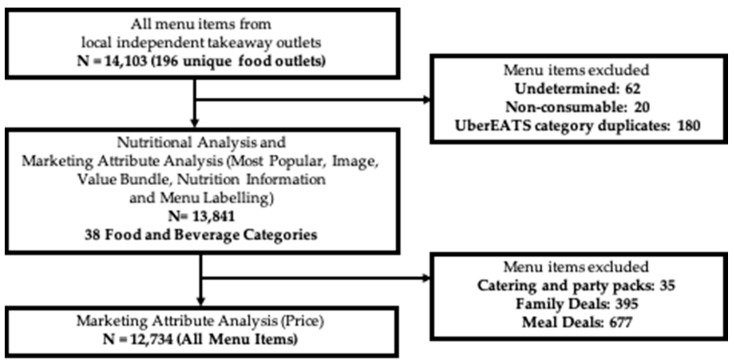
Flow diagram outlining inclusion of menu items in each analysis.

**Figure 2 nutrients-13-00905-f002:**
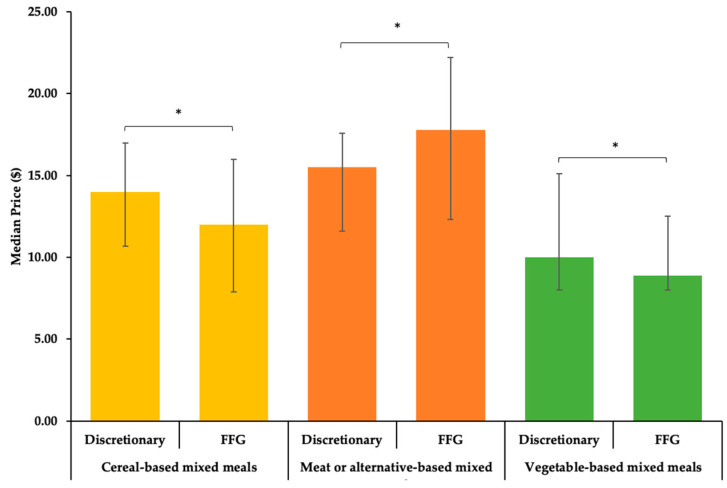
Median price of discretionary and FFG mixed meal categories excluding value bundles. Error bars display interquartile interval, * *p* < 0.02.

**Table 1 nutrients-13-00905-t001:** Summary of definitions and derivations of data extracted from complete menus of each independent takeaway and study outcomes.

Data Extracted	Definition
Menu item name	The name of menu items from a food outlet’s webpage
Menu item description	The description provided for menu items from a food outlet’s webpage. This description is located below the menu item name. Not all menu items have descriptions.
UberEats category	The menu category, which menu items are grouped within on a food outlet’s webpage (e.g., Beverages, Main Meals).
Catering and party packs	Any menu items with “catering”, “party”, or similar terms in either the UberEats category or the menu item name. These menu items were suspected to serve more than 10 people.
UberEats category duplicate	The duplicate menu items that varied only by the Uber- Eats category. These menu items were listed both as “Most Popular” and as another UberEats category (e.g., Chicken Burger listed under Most Popular and Burger categories).
Meal deal	Any menu item that included multiple food components which could be purchased individually from the food outlet (e.g., burger with chips and drink). These menu items were available at a reduced price compared to buying the individual components separately. This was determined using the menu item description in the context of the food outlet’s complete menu.
Family deal	Any menu item intended to serve more than one person and suspected to serve less than ten people. These items contained the terms “for two”, “for three”, “family”, or similar in the UberEATS category, menu name or description.
**Study Outcomes**	**Definition**
Discretionary food or beverage	Foods and beverages, which are defined as items high in added salt, saturated fat, sugar, and low in fibre by the Australian Dietary Guidelines [10,13]. Internationally they are also referred to as junk food or non-core.
Five Food Group (FFG) food or beverage	Foods and beverages, which have food(s)the or combination of foods from the five food groups defined by the Australian Dietary Guidelines: vegetables and legumes/beans, fruit, grain (cereal) foods, mostly wholegrain and/or high cereal fibre varieties; lean meats and poultry, fish, eggs, tofu, nuts and seeds and legumes/beans; and milk, yoghurt cheese, and/or alternatives, mostly reduced fat [10,13] Internationally these are also referred to as core foods.
Most popular menu items	Menu items listed as “Most Popular” in the UberEats category. These items are typically positioned at the top of a food outlet’s UberEats webpage and app interface, attracting greater visibility. All other menu items are referred to as regular menu items.
Value bundles	This is a collective term for meal deals, family deals and catering and party packs.
Image	The image accompanying the menu item name, description, and price. Not all menu items have images.
Price ($)	The price of menu items from a food outlet’s webpage.
Nutritional information	Any information provided on the OFD platform that quantifies any macronutrient(s) of a menu item (e.g., energy, protein) or micronutrient(s) (e.g., sodium). Not all menu items have nutritional information.
Dietary labelling	Any menu item label associated with a dietary requirement (e.g., vegan). Religious dietary labelling (e.g., halal) and heat scale labelling (e.g., spicy) was excluded from this data. Not all menu items have dietary labelling.

**Table 2 nutrients-13-00905-t002:** The proportion of food and beverage categories in complete menus (*N* = 13,841) from 196 independent takeaways. Categories are sorted in descending order.

Type of Category	Food Categories	*n*	%
**Discretionary**	Cereal-Based Mixed Meal	5849	42.3
	Meat Or Alternative Based Mixed Meal	1924	13.9
	Sugar Sweetened Beverages	776	5.6
	Savoury Sauces, Condiments And Spreads	587	4.2
	Fried Potato (Or Similar)	419	3.0
	Baked Goods/Desserts (Homemade Or Similar)	402	2.9
	Vegetable-Mased Mixed Meal	347	2.5
	Other Beverage ^b^	318	2.3
	Iced Confectionary And Dairy-Based Desserts	197	1.4
	Discretionary Milk Based Beverages	194	1.4
	Other Food ^a^	126	0.9
**FFG**	Cereal-Based Mixed Meal	785	5.7
	Vegetable-Based Mixed Meal	583	4.2
	Meat Or Alternative Based Mixed Meal	475	3.4
	Water	248	1.8
	Other Food ^c^	238	1.7
	Other Beverage ^d^	211	1.5
	Juice	162	1.2
	**Total**	13,841	

^a^ Confectionery, Discretionary Snack Food (Savoury)—Packaged, Discretionary Snack Food (Sweet)—Packaged, Other Snack Food (Other), Processed Meats, ^b^ Alcohol, Energy Drinks, Non-Sugar Sweetened Beverages, Rehydration Beverages (Electrolytes), Water Based Flavoured Beverage—Sugar Not Determined, ^c^ Breads And Cereals, Dairy And Alternatives, Fats/Oils, Fruit, Legumes, Meat And Alternatives, Soup, Vegetables, Vegetables (Other), ^d^ Body Building And Performance Beverages, Coffee, Milk/Milk Alternatives, Milk/Milk Alternative Based Beverages, Tea.

**Table 3 nutrients-13-00905-t003:** The proportion of discretionary menu items compared against FFG menu items within marketing attribute subgroups and complete menus ^1^.

Marketing Attribute	Discretionary (%)	FFG (%)	Total	Odds Ratio (95% CI)
Most Popular	568 (90.8)	57 (9.2)	625	2.5 (1.9–3.2)
Image	2419 (59.0)	687 (41.0)	4097	1.3 (1.2–1.5)
Value Bundle	1064 (96.1)	43 (3.9)	1107	6.5 (4.8–8.9)
Complete Menus	11,139 (80.5%)	2702 (19.5%)	13,841	

^1^ The odds ratio was calculated for discretionary categories compared against FFG categories. Percentages are within each marketing attribute.

**Table 4 nutrients-13-00905-t004:** Prevalence of images and value bundles for complete menus (*N* = 13,841) and most popular menu items (*n* = 625) of 196 independent takeaways ^1^.

Food & Beverage Group	Food & Beverage Category	Marketing Attributes	Complete Menus		Most Popular Menu Items	
			*n*	%	*n*	%
**Food (discretionary)**	Cereal-Based Mixed Meal	Image	1846	31.6	318	89.1 *
		Value Bundle	729	12.5	28	7.8 *
	Meat Or Alternative-Based Mixed Meal	Image	680	35.3	137	87.3 *
		Value Bundle	304	15.8	24	15.3
	Savoury Sauces, Condiments And Spreads	Image	87	14.8	0	0
		Value Bundle	2	0.3	0	0
	Fried Potato (Or Similar)	Image	146	34.8	14	87.5 *
		Value Bundle	14	3.3	0	0
	Baked Goods/Desserts (Homemade Or Similar)	Image	149	37.1	8	100
		Value Bundle	2	0.5	0	0
	Iced Confectionary And Dairy-Based Desserts	Image	69	35.0	3	100
		Value Bundle	0	0	0	0
	Vegetable-Based Mixed Meal	Image	133	38.3	18	90.0 *
		Value Bundle	12	3.5	1	5.0
	Other Food ^a^	Image	35	27.8	2	66.7
		Value Bundle	1	0.8	0	0
**Beverage (discretionary)**	Sugar Sweetened Beverages	Image	177	22.8	0	0
		Value Bundle	0	0	0	0
	Other Beverage ^b^	Image	60	18.9	0	0
		Value Bundle	0	0	0	0
	Milk Based Beverages	Image	38	19.6	1	50.0
		Value Bundle	0	0	0	0
**Total Discretionary**		Image	3420	30.7 **	501	88.2 *
		Value Bundle	1064	9.6 **	53	9.3
**Food (FFG)**	Cereal-Based Mixed Meal	Image	166	21.1	20	90.9 *
		Value Bundle	18	2.3	0	0
	Vegetable-Based Mixed Meal	Image	226	38.8	14	93.3 *
		Value Bundle	9	1.5	0	0
	Meat Or Alternative-Based Mixed Meal	Image	129	27.2	15	78.9 *
		Value Bundle	14	2.9	0	0
	Other Food ^c^	Image	41	17.2	1	100
		Value Bundle	2	0.8	0	0
**Beverage (FFG)**	Water	Image	69	27.8	0	0
		Value Bundle	0	0	0	0
	Other Beverage ^d^	Image	20	9.5	0	0
		Value Bundle	0	0	0	0
	Juice	Image	26	16.0	0	0
		Value Bundle	0	0	0	0
**Total FFG**		Image	677	25.1	50	87.7 *
		Value Bundle	43	1.6	0	0
**Total**		Image	4097	29.6	551	88.2 *
		Value Bundle	1107	8.0	53	8.5

^a^ Confectionery, Discretionary Snack Food (Savoury)—Packaged, Discretionary Snack Food (Sweet)—Packaged, Other Snack Food (Other), Processed Meats. ^b^ Alcohol, Energy Drinks, Non-Sugar Sweetened Beverages, Rehydration Beverages (Electrolytes), Water Based Flavoured Beverage—Sugar Not Determined. ^c^ Breads And Cereals, Dairy And Alternatives, Fats/Oils, Fruit, Legumes, Meat And Alternatives, Soup, Vegetables, Vegetables (Other). ^d^ Body Building And Performance Beverages, Coffee, Milk/Milk Alternatives, Milk/Milk Alternative Based Beverages, Tea. ^1^ Percentages are within each Food & Beverage Category where displayed, otherwise within the Total. * *p* < 0.01 compared to regular menu items. ** *p* <0.001 compared to Total FFG (Complete Menus).

**Table 5 nutrients-13-00905-t005:** Median price of most popular and regular menu items for each food or beverage category. Value bundles were excluded (N = 12,734).

Food or Beverage Group	Food or Beverage Category	Most Popular			Regular			*p*-Value
		Median Price ($)	Q1	Q3	Median Price ($)	Q1	Q3	
**Food (Discretionary)**	Cereal-Based Mixed Meal	14.5	12.00	17.00	14.00	10.50	17.00	0.011 *
	Meat Or Alternative-Based Mixed Meal	17.90	14.00	22.95	15.00	10.00	19.95	0.00 *
	Savoury Sauces, Condiments And Spreads	11.50	6.99	16.00	2.00	1.30	3.00	0.025 *
	Fried Potato (Or Similar)	7.95	6.25	8.60	6.80	5.00	8.50	0.195
	Baked Goods/Desserts (Homemade Or Similar)	8.75	4.50	11.00	7.00	5.00	9.98	0.709
	Vegetable-Based Mixed Meal	15.00	10.90	16.95	9.90	7.70	15.00	0.021 *
	Iced Confectionary And Dairy-Based Desserts	5.00	2.50	5.50	6.75	5.50	10.00	0.036 *
	Other Food ^a^	8.00	7.00	9.95	5.93	3.95	10.00	0.305
**Food (FFG)**	Cereal-Based Mixed Meal	12.48	9.95	15.90	12.00	7.90	16.00	0.428
	Vegetable-Based Mixed Meal	12.00	8.50	13.00	8.90	8.00	12.50	0.066
	Meat Or Alternative-Based Mixed Meal	18.00	15.95	21.40	17.80	13.90	19.90	0.331
	Other Food ^c^	6.00	3.00	8.90	8.80	8.80	8.80	0.452
**Beverage (Discretionary)**	Sugar Sweetened Beverages	-	-	-	4.40	3.50	5.50	-
	Other Beverage ^b^	-	-	-	4.75	4.00	5.50	-
	Discretionary Milk Based Beverages	8.45	7.90	9.00	7.50	5.00	8.50	0.384
**Beverage (FFG)**	Water	-	-	-	3.90	3.00	4.90	-
	Other Beverage ^d^	-	-	-	4.50	4.00	6.00	-
	Juice	-	-	-	4.90	4.00	5.50	-

^a^ Confectionery, Discretionary Snack Food (Savoury)—Packaged, Discretionary Snack Food (Sweet)—Packaged, Other Snack Food (Other), Processed Meats. ^b^ Alcohol, Energy Drinks, Non-Sugar Sweetened Beverages, Rehydration Beverages (Electrolytes), Water Based Flavoured Beverage—Sugar Not Determined. ^c^ Breads And Cereals, Dairy And Alternatives, Fats/Oils, Fruit, Legumes, Meat And Alternatives, Soup, Vegetables, Vegetables (Other). ^d^ Body Building And Performance Beverages, Coffee, Milk/Milk Alternatives, Milk/Milk Alternative Based Beverages, Tea. * *p* < 0.05. Q = Quartile.

## Data Availability

Publicly available datasets were analyzed in this study. This data can be found here: ubereats.com (accessed on September 10th 2020).

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
