# Peer review of "Hunger for Home Delivery: Cross-Sectional Analysis of the Nutritional Quality of Complete Menus on an Online Food Delivery Platform in Australia"

_nutrients, 2021, doi:10.3390/nu13030905_

Round 1

Reviewer 1 Report

I now read the "Hunger for Home Delivery" manuscript submitted to Nutrients. It reports on a content analysis of the foods UberEats offers in the Sydney region. Given the rising popularity of this type of services, the manuscript deals with a timely and relevant topic, which extends previous knowledge on related consumption choices. The research and findings are interesting, but I still think that it should be more strongly embedded in the food marketing research that it links to. At present, the paper almost seems to be written in vacuum when it comes to this food marketing literature, while it would be extremely relevant to take that literature into account. As such, it would be more clear what type of findings transfer from other consumption choice settings, and which trends even grow bigger in the online food delivery platform. Moreover, the method description needs some further clarification to make the research replicable (as has also been done in other food marketing areas) and to make the findings more interpretable for the reader. Below, I address these points in more detail.

1. There have been lots of studies on supermarket offering. Granted, many of these focused on foods targeting children, but the content analysis (both in terms of healthiness and in terms of marketing cues) of these studies offers crucial insights in both the methodology and the findings. Both methods and findings of these studies offer a good backdrop for the present manuscript. 

Examples of references:

-Aerts, G., & Smits, T. (2019). Child-targeted on-pack communications in Belgian supermarkets: Associations with nutritional value and type of brand. Health promotion international34(1), 71-81.

-Campbell, S., James, E. L., Stacey, F. G., Bowman, J., Chapman,
K. and Kelly, B. (2014) A mixed-method examination of
food marketing directed towards children in Australian supermarkets.
Health Promotion International, 29, 267–277.

-Chapman, K., Nicholas, P., Banovic, D. and Supramaniam, R.
(2006) The extent and nature of food promotion directed to
children in Australian supermarkets. Health Promotion
International, 21, 331–339.
-Elliott, C. D. (2008) Assessing 0fun foods0: nutritional content
and analysis of supermarket foods targeted at children.
Obesity Review, 9, 368–378.

-Hallez, L., Qutteina, Y., Raedschelders, M., Boen, F., & Smits, T. (2020). That’s my cue to eat: A systematic review of the persuasiveness of front-of-pack cues on food packages for children vs. adults. Nutrients12(4), 1062.

-Harris, J. L., Schwartz, M. B. and Brownell, K. D. (2010)
Marketing foods to children and adolescents: licensed characters
and other promotions on packaged foods in the supermarket.
Public Health Nutrition, 13, 409–417.

-Van Assema, P., Joosten, S., Bessems, K., Raaijmakers, L., de
Vries, N. and Kremers, S. (2011) De omvang en aard van
verkoopstrategiee¨n gericht op kinderen bij voedingsmiddelen.
Tijdschrift Voor Gezondheidswetenschappen, 89,
108–113.

References to content analyses on particular media:

-Alvy, L. M. and Calvert, S. L. (2008) Food marketing on popular
children’s web sites: a content analysis. Journal of the
American Dietetic Association, 108, 710–713.

-Neyens, E. and Smits, T. (2016) Empty pledges: a content analysis
of Belgian and Dutch child-targeting food websites.
International Journal of Health Promotion and Education,
55, 1–11.

-Qutteina, Y., Hallez, L., Mennes, N., De Backer, C., & Smits, T. (2019). What do adolescents see on social media? A diary study of food marketing images on social media. Frontiers in psychology10, 2637.

2. Another type of literature that I think is interesting to relate to deals with the persuasive cues on online webshops. For example:

Breugelmans, E., Campo, K. and Gijsbrechts, Els. (2007) Shelf
sequence and proximity effects on online grocery choices.
Marketing Letters, 18(Suppl. 1), 117–133.

3. With regard to COVID (line 54), I suggest to have a look at this recent article: https://doi.org/10.3389/fnut.2020.621726

4. With regard to the sampling of the meals, I would like to see some more information. Are the authors sure that they saw the objectively most popular menu items (double check with anonymous browsers etc.), or could this have been personalized results? How exactly was the "near you" operationalized?

5. It was unclear to me whether the data comprised the 680 unique outlets or the 202 independent ones. The manuscript seems to be going back and forth a bit with regard to this. Moreover, can we get some info on the non-independent outlets then?

6. The manuscript uses quite some terminology specific to Australia (e.g. discretionary foods, FFG, ...) and this might need more explanation for the other readers. 

7. Although the article is positioned as a content analysis of both the healthiness and the marketing techniques of the online platform's offering, the marketing-related variables are quite superficial. Many more characterics could be derived (for ideas: see also the references above from the previous research on food marketing and online grocery shopping). Although I do understand that not all of these aspects can be taken into account at this moment, I do suggest to drastically increase the marketing focus or to drop it all together (which would make the manuscript a lot less interesting, in my opinion). But in its current form, the manuscript is promising a value bundle, but does not deliver on that promise.

7. Section 2.4 was hard to follow. In line with this, I actually found the whole results section hard to follow. For instance, it took me quite some puzzling to understand Table 2. The reader has to figure a lot out based on the Tables and text, while I think this is a job that the authors should better facilitate this. This might also imply that some of the tables move to additional materials and that the authors try to present these findings in a more systematic and insightful way themselves. Some of the wording also is confusing: for instance in section 3.2.3 one reads that discretionary items had more images, but I guess this is "are more likely to have an image"?  Please also provide more statistics such that we can understand what test the p-values actually refer to.

I wish the authors good luck with their interesting research!

Author Response

Reviewer 1

  1. I now read the "Hunger for Home Delivery" manuscript submitted to Nutrients. It reports on a content analysis of the foods UberEats offers in the Sydney region. Given the rising popularity of this type of services, the manuscript deals with a timely and relevant topic, which extends previous knowledge on related consumption choices. The research and findings are interesting, but I still think that it should be more strongly embedded in the food marketing research that it links to. At present, the paper almost seems to be written in vacuum when it comes to this food marketing literature, while it would be extremely relevant to take that literature into account. As such, it would be more clear what type of findings transfer from other consumption choice settings, and which trends even grow bigger in the online food delivery platform. Moreover, the method description needs some further clarification to make the research replicable (as has also been done in other food marketing areas) and to make the findings more interpretable for the reader. Below, I address these points in more detail.

Our response: Thank you for your review of our paper. We have taken on board your suggestions and we believe this has markedly strengthened our manuscript. Please refer to our response to comment 2 and 3 regarding adding additional food marketing literature. Please refer to the response to comment 9 in regard to clarifying methodology.

  1. There have been lots of studies on supermarket offering. Granted, many of these focused on foods targeting children, but the content analysis (both in terms of healthiness and in terms of marketing cues) of these studies offers crucial insights in both the methodology and the findings. Both methods and findings of these studies offer a good backdrop for the present manuscript.

Examples of references:

  • Aerts, G., & Smits, T. (2019). Child-targeted on-pack communications in Belgian supermarkets: Associations with nutritional value and type of brand. Health promotion international, 34(1), 71-81.
  • Campbell, S., James, E. L., Stacey, F. G., Bowman, J., Chapman, K. and Kelly, B. (2014) A mixed-method examination of food marketing directed towards children in Australian supermarkets. Health Promotion International, 29, 267–277.
  • Chapman, K., Nicholas, P., Banovic, D. and Supramaniam, R.(2006) The extent and nature of food promotion directed to children in Australian supermarkets. Health Promotion International, 21, 331–339.
  • Elliott, C. D. (2008) Assessing 0fun foods0: nutritional content and analysis of supermarket foods targeted at children. Obesity Review, 9, 368–378.
  • Hallez, L., Qutteina, Y., Raedschelders, M., Boen, F., & Smits, T. (2020). That’s my cue to eat: A systematic review of the persuasiveness of front-of-pack cues on food packages for children vs. adults. Nutrients, 12(4), 1062.
  • Harris, J. L., Schwartz, M. B. and Brownell, K. D. (2010) Marketing foods to children and adolescents: licensed characters and other promotions on packaged foods in the supermarket. Public Health Nutrition, 13, 409–417.
  • Van Assema, P., Joosten, S., Bessems, K., Raaijmakers, L., de Vries, N. and Kremers, S. (2011) De omvang en aard van verkoopstrategiee¨n gericht op kinderen bij voedingsmiddelen. Tijdschrift Voor Gezondheidswetenschappen, 89, 108–113.

References to content analyses on particular media:

  • Alvy, L. M. and Calvert, S. L. (2008) Food marketing on popular children’s web sites: a content analysis. Journal of the American Dietetic Association, 108, 710–713.
  • Neyens, E. and Smits, T. (2016) Empty pledges: a content analysis of Belgian and Dutch child-targeting food websites. International Journal of Health Promotion and Education, 55, 1–11.
  • Qutteina, Y., Hallez, L., Mennes, N., De Backer, C., & Smits, T. (2019). What do adolescents see on social media? A diary study of food marketing images on social media. Frontiers in psychology, 10, 2637.

Our response:  We agree with your suggestion to incorporate further marketing attributes literature and have made the below additions in the Introduction and Discussion. The majority of offerings we have investigated are not packaged food items therefore some of the attributes proposed in the references suggested are not observed within our sample. While cross-promotions with outside partners are relevant in the broad advertising of OFD platforms, this is not seen for specific menu items we evaluated. Purchase incentives such as discounts and scarcity cues exist for specific menu items on these platforms, similarly to supermarkets as described in the above references highlighted by the Reviewer. However, these are only visible on platform users’ accounts and this study used publicly available data extracted from the UberEats website (not a personal user account). Additionally, these marketing tactics are often targeted at specific users for a limited time while the data we extracted was of a singular time-point. Consequently, we have addressed the absence of these marketing attributes from our study within the limitations section:

“Marketing tactics that position foods to be first viewed and default popular choices may encourage their selection. Within an online grocery store context, options placed on the ‘first-screen’ appear to be more commonly selected [21]. Such strategies are akin to the posi-tioning of food items on supermarket shelves at eye-level and at check-out counters [22].” Page 3, lines 107-112)

“Nutritional labelling at point-of-sale may also serve as a form of marketing to encourage choice, or in this case to guide consumers to healthier choices [25-27].” Page 3, lines 116-118)

“The use of imagery to market poor nutritional quality food has also been noted on other digital platforms, with a recent study reporting 67% of food images seen by adolescents on social media were of dis-cretionary items [44].” (Page 13, lines 431-434)

Previous research examining the relationship of nutritional quality and food marketing within the digital food environment has focused on online grocery shopping [21].” (Page 14, lines 482-484)

“As we used web scraping for data extraction, we were limited to the marketing attributes available through the webbrowser interface and not the mobile application interface. We also could not examine pro-motions and purchase incentives which are only visible on users’ personal accounts. Our findings were interpreted with the assumption that ‘Most Popular’ referred to items with the greatest sales, however the algorithm used to determine this characteristic is not made publicly available.” (Pages 15, lines 499-506)

  1. Another type of literature that I think is interesting to relate to deals with the persuasive cues on online webshops. For example:

Breugelmans, E., Campo, K. and Gijsbrechts, Els. (2007) Shelf sequence and proximity effects on online grocery choices. Marketing Letters, 18(Suppl. 1), 117–133.

Our response: We have incorporated this research into our introduction.

“Within an online grocery store context, options placed on the ‘first-screen’ appear to be more commonly selected [21].” (Page 3, lines 108-110)

  1. With regard to COVID (line 54), I suggest to have a look at this recent article: https://doi.org/10.3389/fnut.2020.621726

Our response: We have moved comments regarding COVID-19 to the Discussion and have added this.

“It is possible that the COVID-19 pandemic may have increased the ordering of unhealthy choices on OFD platform. A recent observational study in 38 countries reported COVID-19 stay at home policies and COVID-19 induced psychological distress was associated with less preparing and selecting of healthier foods in adults [35]. This is high-lighted as UBER reported that compared to the previous year, delivery bookings grew 113% in the second quarter of 2020 and revenue grew 103% in August 2020 [36].” (Pages 12-13, lines 384-392)

  1. With regard to the sampling of the meals, I would like to see some more information. Are the authors sure that they saw the objectively most popular menu items (double check with anonymous browsers etc.), or could this have been personalized results? How exactly was the "near you" operationalized?

Our response: We have clarified that we did not obtain personalised results. The algorithm for the UberEats’ ‘popular near you’ section is not publicly available. We have rewritten our first paragraph of the methods to summarise relevant detail and guide readers to the previous study for additional details on how the popular food outlets were identified.

“2.1. Identification of popular independent takeaways

This study formed a new database of complete menus for food outlets that were identified by a previous cross-sectional study con-ducted in Sydney between 9-22 February 2020. The identification process is described in detail elsewhere [10]. The 10 most popular food outlets were extracted from the UberEats’ ‘popular near you’ section for areas with above-average populations of young people (>30% 15-34 years), the leading users of OFD platforms [10]. Researchers were not logged into personal UberEats accounts to avoid biased results. This project focused on evaluation of the 202 independent takeaways identified from the previous study using the Food Environment Score Tool [16,29,30]. Independent takeaways have been defined as outlets that are not franchises, which prepare and sell meals or snacks, ready for immediate consumption e.g. local kebab shop [16]. (Page 3, lines 133-146)

  1. It was unclear to me whether the data comprised the 680 unique outlets or the 202 independent ones. The manuscript seems to be going back and forth a bit with regard to this. Moreover, can we get some info on the non-independent outlets then?

Our response: Our methods indicated that we focused on the 202 independent takeaways and the purpose of this was elaborated in our introduction. We agree that reference to the 680 unique outlets may add confusion and thus have removed mention of the 680 outlets from the first section of our Methods, Figure 1 and Section 3.1. Further information on all outlets is available from the previous study: https://doi.org/10.3390/nu12103107

“This project focused on evaluation of the 202 independent takeaways identified from the previous study using the Food Environment Score Tool [16,29,30].” (Page 3 lines 141-144)

  1. The manuscript uses quite some terminology specific to Australia (e.g. discretionary foods, FFG, ...) and this might need more explanation for the other readers.

Our response: Our Introduction includes a definition of discretionary foods; we have added alternative international terminology to this also. We have added a definition for Five Food Groups and discretionary foods in the Section 2.3.1 of the Methods. These definitions have also been added to Table 1 for further clarity.  

“were discretionary foods and beverages (also known as junk food) which are defined as items high in added salt, saturated fat, sugar and low in fibre by the Australian Dietary Guidelines [10,13].” (Page 2 lines 82-85)

“FFG dishes (also known as core) contain food(s) or a combination of foods from the five food groups: vegetables and legumes/beans, fruit, grain (cereal) foods, mostly wholegrain and/or high cereal fibre vari-eties; lean meats and poultry, fish, eggs, tofu, nuts and seeds and legumes/beans; and milk, yoghurt cheese and/or alternatives, mostly reduced fat) [10,13]. Discretionary dishes are defined as items high in added salt, saturated fat, sugar and low in fibre by the Australian Dietary Guidelines [10,13].” (Page 5, lines 171-178)

Please also refer to Table 1 for definitions.

  1. Although the article is positioned as a content analysis of both the healthiness and the marketing techniques of the online platform's offering, the marketing-related variables are quite superficial. Many more characterics could be derived (for ideas: see also the references above from the previous research on food marketing and online grocery shopping). Although I do understand that not all of these aspects can be taken into account at this moment, I do suggest to drastically increase the marketing focus or to drop it all together (which would make the manuscript a lot less interesting, in my opinion). But in its current form, the manuscript is promising a value bundle, but does not deliver on that promise.

Our response: Thank you for your feedback regarding the examination of marketing-related variables. Unfortunately, as a cross-sectional study, we cannot comment on common transient marketing tactics such as price promotions mentioned in the references above. Additionally, these promotions are typically sent to users’ accounts, although we did not extract our data from private accounts. Furthermore, the type of large-scale web scraping utilised by this study can only be conducted on webpages not on mobile applications (which are the preferred interface by users) and these two interfaces vary significantly. The concept of greater visibility of the ‘most popular’ category carries through from the webpages to the application. Exploring other positioning marketing tactics would not offer great value as layout features do not carry through between the interfaces. Similarly, we explored the other marketing attributes (image, offering as meal deal, price, nutritional information, and dietary labelling) that all carry through from the webpage to the application. Given the scarcity of research in OFD platform-specific marketing attributes, our research still adds great value to this emerging field as it currently stands. By examining over 14,000 menu items, to our knowledge the largest data set evaluated within the digital food environment, we are confident in the importance of our findings. These findings will serve as a foundation for future research to build upon, that will explore further marketing tactics and their impact on consumption data to establish more causal relationships between marketing attributes and nutritional quality. Our nutritional analysis component is also of importance. We studied data from independent food outlets, and their nutritional quality has been previously understudied. Our findings highlight the need for their inclusion for public health discussions. 

  1. Section 2.4 was hard to follow. In line with this, I actually found the whole results section hard to follow. For instance, it took me quite some puzzling to understand Table 2. The reader has to figure a lot out based on the Tables and text, while I think this is a job that the authors should better facilitate this. This might also imply that some of the tables move to additional materials and that the authors try to present these findings in a more systematic and insightful way themselves. Some of the wording also is confusing: for instance in section 3.2.3 one reads that discretionary items had more images, but I guess this is "are more likely to have an image"? Please also provide more statistics such that we can understand what test the p-values actually refer to.

Our response: We have amended section 2.4 to include the post-hoc tests used to identify the significant differences between i) non-core and core menu items and ii) most popular and regular menu items. We have changed Table 2 to have the proportion of food and beverage categories for complete menus only and have moved the original Table 2B to the appendix. We also changed the wording for section 3.2.3. Throughout the results, we have added sample sizes to assist the reader and further information to the tables’ footnotes.

“Chi-squared tests with the Bonferroni multiple comparisons correction”(Page 6, lines 215-216)

“Kruskal-Wallis tests with multiple comparisons corrections” (Page 6, lines 221-222)

“Table 3 depicts the proportion of discretionary and FFG menu items within complete menus and each marketing attribute subgroup.” (Page 7, lines 253-254)

“A higher proportion of discretionary menu items (9.6%, 1064/11139) were offered as a value bundle compared to FFG menu items (1.6%, 43/2702) (P<0.001) (Table 4).”(Page 9, lines 283-285)

“A higher proportion of discretionary menu items (30.7%, 3420/11139) had images compared to FFG menu items (25.1%, 677/2702) (P<0.001) (Table 4).” (Page 9, lines 294-296)

Thank you for reviewing this manuscript,

Andriana Korai and Celina Wang on behalf of all authors

Reviewer 2 Report

The paper is interesting because it deals with an extremely timely issue, such as the nutritional quality of all menu from popular independent takeaways, and the associations between the nutritional quality and marketing attributes.

I would only introduce only a few changes.

At the end of the Introduction, I would better specify the articulation of the paper. The theoretical framework should be further developed, including methodological aspects.
The conclusions are too concise, especially regarding the limitations of the work and future lines of research.

The text contains several typos that need to be corrected.

Author Response

Reviewer 2

  1. The paper is interesting because it deals with an extremely timely issue, such as the nutritional quality of all menu from popular independent takeaways, and the associations between the nutritional quality and marketing attributes. I would only introduce only a few changes.

Our response: Thank you for your comments.

  1. At the end of the Introduction, I would better specify the articulation of the paper. The theoretical framework should be further developed, including methodological aspects.

Our response: We have added additional information at the end of the Introduction for more specificity. However, as our research is cross-sectional and we did not explore consumption behaviours, we have not delved into specific theoretical underpinnings for marketing attributes.

“The use of marketing techniques within the unique digital food environment of OFD platforms and the association with nutritional quality of menus offered warrants further investigation.” (Page 3, lines 121-123)

“in areas with high proportions of young consumers (15-34-years) in Australia.” (Page 3, lines 126-127)

  1. The conclusions are too concise, especially regarding the limitations of the work and future lines of research.

Our response: We have added additional limitations and future lines of research. We have edited our concluding line to clarify the future lines of research further.

“Our nutritional analysis was also limited to food and beverage cate-gorisation because we had no access to complete recipe data of all menu items. To obtain a more detailed nutrient analysis of independent outlets’ offerings, more detailed food composition data would be needed.” (Page 15, lines 519-523)

“However, we did not examine consumption nor even sales, thus, the impact remains speculative. We strongly advocate for further research examining how marketing cues in OFD platforms affect nutrition in-takes and determine the need to expand menu labelling policies.” (Page 15, lines 525-529)

  1. The text contains several typos that need to be corrected.

Our response: We have reviewed our manuscript for typos.

Thank You for reviewing this manuscript,

Andriana Korai and Celina Wang on behalf of all authors

Reviewer 3 Report

In their paper, Wang et al present the cross-Sectional Analysis of The Nutritional Quality of Menu Items on an Online Food Delivery platform in Australia. In performing the analysis, the author did an excellent job and I consider the paper an important addition to the literature in the field.  However, I have a problem to be solved: the authors claim to be the first to evaluate the nutritional quality of all menu items offered by popular independent takeaways on an Australian market-leading OFD platform. Unfortunately, I am slightly concerned by the similarity of another article by this team: https://pubmed.ncbi.nlm.nih.gov/33053705/ Could the authors explain the added value of the presented article in comparison to this one? What the core differences are? Obviously, the studies were conducted in the same geographical areas using the same analytics, hence I am not convinced about the novel added value of the paper.

Please, could you provide an explanation of the similarity between these two articles?

Otherwise, I find the article meticulously and properly prepared and would recommend its publication.

Author Response

Reviewer 3

In their paper, Wang et al present the cross-Sectional Analysis of The Nutritional Quality of Menu Items on an Online Food Delivery platform in Australia. In performing the analysis, the author did an excellent job and I consider the paper an important addition to the literature in the field.  However, I have a problem to be solved: the authors claim to be the first to evaluate the nutritional quality of all menu items offered by popular independent takeaways on an Australian market-leading OFD platform. Unfortunately, I am slightly concerned by the similarity of another article by this team: https://pubmed.ncbi.nlm.nih.gov/33053705/ Could the authors explain the added value of the presented article in comparison to this one? What the core differences are? Obviously, the studies were conducted in the same geographical areas using the same analytics, hence I am not convinced about the novel added value of the paper.

  1. Please, could you provide an explanation of the similarity between these two articles? Otherwise, I find the article meticulously and properly prepared and would recommend its publication.

Our response: The original work identified and characterised the food outlets explored in the current study and evaluated the nutritional quality of their most popular menu items only (936 menu items from the same 202 outlets) using a binary classification (discretionary or core). The current study evaluated the nutritional quality of the complete menus of these 202 outlets, a sample of 14,021 menu items, using a novel classification system of 38 food and beverage categories. As this study focused on a single type of food outlet (independent takeaways) this more comprehensive classification system allowed a thorough characterisation of offerings from this food outlet type. Furthermore, this study also explored how novel platform-related marketing attributes are related to the nutritional quality of the menu items (popularity cue, use of image and price) and conducted a similar analysis of meal deals on the complete sample (n=14,021) not only the most popular menu items. We understand there may have been some confusion to the reader and have amended our title to include ‘complete menus’. We have also adjusted the methods to clarify the difference further by changing our first paragraph of the methods and added additional content to the discussion to highlight the added value of the current study.

“2.1. Identification of popular independent takeaways

This study formed a new database of complete menus for food outlets that were identified by a previous cross-sectional study con-ducted in Sydney between 9-22 February 2020. The identification process is described in detail elsewhere [10]. The 10 most popular food outlets were extracted from the UberEats’ ‘popular near you’ section for areas with above-average populations of young people (>30% 15-34 years), the leading users of OFD platforms [10]. Researchers were not logged into personal UberEats accounts to avoid biased results. This project focused on evaluation of the 202 independent takeaways identified from the previous study using the Food Environment Score Tool [16,29,30]. Independent takeaways have been defined as outlets that are not franchises, which prepare and sell meals or snacks, ready for immediate consumption e.g. local kebab shop [16]. (Page 3, lines 133-146)

“To our knowledge this is the first study to link the nutritional quality and food marketing attributes for food prepared outside the home within the unique digital environment of OFD platforms.” (Page 14, 484-487)

Thank You for reviewing this manuscript,

Andriana Korai and Celina Wang on behalf of all authors

Round 2

Reviewer 1 Report

I wish to thank the authors for their detailed response to my comments. I also believe that their revision of the manuscript drastically improved the quality of the content and presentation. I have no further comments and wish the authors good luck with their future research.